# Approaching enzymatic catalysis with zeolites or how to select one reaction mechanism competing with others

Pau Ferri[1,4], Chengeng Li [1,4], Daniel Schwalbe-Koda[2], Mingrou Xie[3], Manuel Moliner [1], Rafael Gómez-Bombarelli [2], Mercedes Boronat [1]✉ & Avelino Corma [1]✉

Approaching the level of molecular recognition of enzymes with solid catalysts is a challenging goal, achieved in this work for the competing transalkylation and disproportionation of diethylbenzene catalyzed by acid zeolites. The key diaryl intermediates for the two competing reactions only differ in the number of ethyl substituents in the aromatic rings, and therefore finding a selective zeolite able to recognize this subtle difference requires an accurate balance of the stabilization of reaction intermediates and transition states inside the zeolite microporous voids. In this work we present a computational methodology that, by combining a fast high-throughput screeening of all zeolite structures able to stabilize the key intermediates with a more computationally demanding mechanistic study only on the most promising candidates, guides the selection of the zeolite structures to be synthesized. The methodology presented is validated experimentally and allows to go beyond the conventional criteria of zeolite shape-selectivity.

Zeolites are microporous aluminosilicates largely employed as efficient heterogeneous catalysts. The presence of isolated tetrahedrally-coordinated $Al^{3+}$ species in the framework generates negative charges that, when compensated with protons, give rise to Brønsted acid properties. This acidity combined with shape-selectivity, and thermal and hydrothermal stability, have made zeolites the most widely used acid catalysts in industry. In addition, their microporous structure of channels and cavities of molecular dimensions with well-determined topologies provide very specific environments for the active sites, leading to an enhanced reactivity associated with confinement effects[1–4]. The stabilization of reaction intermediates and transition states inside the zeolite microporous voids by weak Van der Waals interactions is very sensitive to the fit of the adsorbed species with the surrounding environment, so subtle differences in the zeolite structure or the reactant molecules may lead to significant differences in catalytic performance, in a similar way as it occurs in enzymatic

catalysis. Indeed, attempts to mimic enzymatic behavior with zeolites by placing the active sites in particular positions or voids have abundant precedent[5–12].

However, the way of action of enzymes cannot be fully mimicked by solid - and therefore rigid - materials. Nature has achieved the high reaction selectivity of enzymes by means of an evolutionary process that has made them very specific for the substrate. The active sites in enzymes are placed in spatial pockets or clefts where proximity effects and an adequate preorganization of the reactants within the microenvironment decrease the activation energy by means of weak interactions with the transition state and consequently increase the rate of the desired reaction. Two generic models, the lock and key and the induced fit models are used to explain the specificity of the enzymes[13]. The first one assumes that the enzyme and the substrate possess specific complementary geometric shapes that fit exactly into one another, while the second one suggests that the enzyme conformation

[1]Instituto de Tecnología Química, Universitat Politècnica de València - Consejo Superior de Investigaciones Científicas, Avenida de los Naranjos s/n, 46022 Valencia, Spain. [2]Department of Materials Science and Engineering, Massachusetts Institute of Technology, Cambridge, MA 02139, USA. [3]Department of Chemical Engineering, Massachusetts Institute of Technology, Cambridge, MA 02139, USA. [4]These authors contributed equally: Pau Ferri, Chengeng Li. ✉e-mail: boronat@itq.upv.es; acorma@itq.upv.es

changes continuously when approaching the substrate until the last one is completely bound to what can be considered the active site within the enzyme pocket. It appears that such dynamic behavior favored by the high flexibility of the enzymes could be approached with transition metal complexes and organocatalysts, both having demonstrated extraordinary selectivity for a large variety of reactions. In the case of solid catalysts, a possibility to approach enzymatic behavior could rely on the synthesis of somehow flexible hybrid organic-inorganic materials as for instance metal-organic frameworks (MOFs), mesoporous organic-inorganic materials, or structured but flexible organic microporous materials (PAFs). Positive results have been reported by means of molecular imprinting of mimics of reaction transition states on solids such as silica or polymers, but the removal of the imprinted molecules from the "flexible" solid support often produces changes in the imprinted system that decrease the effective fitting with the transition state and therefore on its stabilization[14-17].

The ability of enzymes to adopt multiple conformations along the catalytic process is missing in solid catalysts like zeolites which have very limited flexibility, and therefore an optimum fitting between the target transition state and the almost rigid crystalline structure of the zeolite is initially required. A recently reported methodology to achieve this fitting during the zeolite synthesis is based on the use of organic structure directing agent (OSDA) molecules that mimic the geometry and charge distribution of the key transition states of the desired chemical reaction. The OSDA-mimic approach has been successfully applied to synthesize zeolites exhibiting improved catalytic performance in selected reactions such as toluene disproportionation[18], ethylbenzene and endo-tricyclodecane isomerizations[18], Diels-Alder cycloaddition[11], and the methanol-to-olefins (MTO) reaction where it was possible to control the selectivity to ethene and propene by favoring one out of two competing reaction mechanisms[19].

With these results in hand, the question now becomes how sensitive this methodology is to discern small variations in composition and structure of the transition state and therefore to differentiate close reaction mechanisms that result in different product selectivities, in a clear parallelism with the enzymatic models. To elaborate on that, we have combined computational and experimental techniques to study a paradigmatic industrial reaction catalyzed by acid zeolites and used as model reaction by the International Zeolite Association, the transalkylation of diethylbenzene with benzene. Diethylbenzene (DEB) is an undesired by-product in the industrial production of ethylbenzene (EB) by alkylation of benzene with ethene and, in order to improve the selectivity of the process, DEB is subsequently converted into EB by reaction with benzene in an additional transalkylation reactor. However, another competitive reaction may take place during DEB transalkylation, the disproportionation of DEB into EB and triethylbenzene (TEB) (Supplementary Fig. 1).

From a mechanistic point of view, each of the two DEB reactions, transalkylation and disproportionation, can proceed through the same two pathways: an alkyl-transfer route involving consecutive deal-kylation and alkylation steps, and a diaryl-mediated pathway involving the formation of bulky cationic diaryl intermediates (Fig. 1)[20-24]. The former route is less selective because the ethoxy intermediates not only can react with all the aromatics present yielding a mixture of alkylated and poly-alkylated products, but they can also undergo dealkylation yielding ethene as byproduct.

Therefore, promoting the diaryl-mediated pathway is the first goal that was achieved applying the OSDA-mimics approach to the DEB transalkylation. Zeolite ITQ-27 with the IWV crystallographic structure was synthesized using as OSDA the diphenyldimehylphosphonium (DMDPP$^+$) cation that contains two aromatic rings connected by a positively charged phosphonium center (Fig. 1c)[18,24]. The geometry and charge localization is similar in the OSDA and in the key diaryl inter-mediates for transalkylation (I$_{trans}$) and disproportionation (I$_{disp}$), and

the only difference among them is the number of ethyl substituents in the aromatic rings. To further improve the catalyst efficiency by sup-pressing the disproportionation of DEB while enhancing transalkyla-tion, the zeolite structure should be able to recognize this subtle difference, approaching the level of molecular recognition of enzymes.

In this work, we use a combination of computational and experi-mental techniques to achieve this challenging goal. Starting with a high-throughput screening of all zeolite structures potentially well-suited to stabilize diaryl intermediates, and improving the accuracy of the thermodynamic and kinetic data by means of periodic DFT calcu-lations, a set of large pore zeolites with different microporous struc-tures of channels and cavities is proposed. The synthesis, characterization, and catalytic test of these materials confirm the theoretical predictions and demonstrate that subtle changes in pore size and architecture can tune the preferred reaction pathways, lead-ing to a precise control of the host-guest interactions that approach the specificity of an enzymatic catalyst.

## Results

### Fast computational screening of potentially efficient zeolites

In a recent work[25] some of us presented a general approach based on high-throughput simulations of more than half a million zeolite-OSDA pairs and on the definition of new design metrics, which led to the phase-selective synthesis of targeted zeolites using newly identified OSDAs, and generated an online database called Organic Structure directing agent DataBase (OSDB) aimed to accelerate the design and optimization of OSDAs for selected zeolites. In this work, we used OSDB to select fifty zeolite structures that better stabilize the DPDMP$^+$ cation (Fig. 1c), which is the OSDA mimicking the diaryl intermediates involved in aromatics transalkylation reactions[18,24]. To evaluate the influence of additional ethyl substituents in the aromatic rings on the stabilization by confinement of diaryl intermediates, the binding energy (BE) between each zeolite structure and the neutral diaryl intermediates for DEB-Bz transalkylation (I$_{trans}$, Fig. 1c) and for DEB disproportionation (I$_{disp}$, Fig. 1c) was evaluated using force fields (FF) as described in Methods Section. One organic molecule was intro-duced in each zeolite unit cell in different initial positions, and only the BE for the most stable location after optimization or each zeolite-intermediate pair are listed in Supplementary Table 1 and plotted in Supplementary Fig. 2. As expected, there is a clear correlation between BE I$_{trans}$ and BE$_{OSDA}$ values, which supports the transition state mimics approach for OSDAs selection and a slightly worse correlation between BE I$_{disp}$ and BE$_{OSDA}$ values due to the presence of additional ethyl groups in both aromatic rings. The plot of BE I$_{disp}$ versus BE I$_{trans}$ suggests that some zeolite structures might be able to discriminate between transalkylation and disproportionation intermediates (Sup-plementary Fig. 2b).

The fifty zeolites in this set exhibit very diverse internal micro-porous structures, including closed cavities only accessible through small 8-ring windows, linear channels of different diameters running parallel along the crystal, and bi-directional (2D) or three-directional (3D) systems of channels that intersect generating wider internal voids. Based on our previous knowledge of the reaction, we must exclude from this set cage-based small-pore zeolites which do not allow the diffusion of aromatic reactants and products through the 8-ring win-dows (blue points in Supplementary Fig. 2b), and zeolites with a 1D channel system that is easily blocked by diaryl intermediates leading to fast catalyst deactivation (orange points in Supplementary Fig. 2b). In zeolites containing 2D 10-ring channel systems (yellow points in Sup-plementary Fig. 2b) the alkyl-transfer pathway is preferred, and although diaryl intermediates might form at the channels intersection they cannot react properly, again leading to catalyst blocking and deactivation[23,24]. With these criteria, the set of potential zeolite candi-dates for DEB-Bz transalkylation drops to eleven (red points in Sup-plementary Fig. 2b), summarized in Table 1. Notice that the most

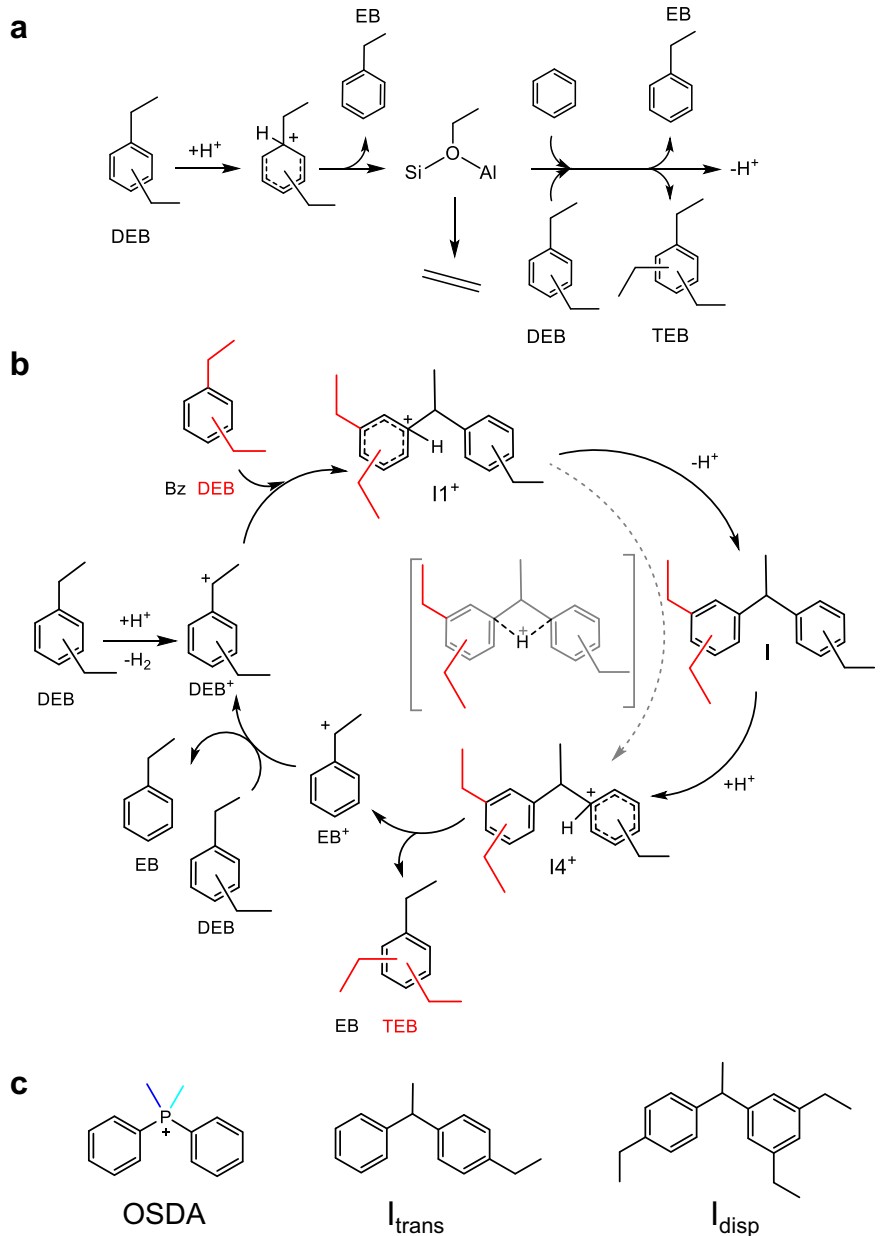

**Fig. 1 | Mechanistic pathways for Diethylbenzene transformations. a** Alkyl-transfer pathway for transalkylation of diethylbenzene with benzene and competitive dealkylation yielding ethene and disproportionation producing triethylbenzene. **b** Diaryl-mediated pathway for the transalkylation of diethylbenzene with benzene and for the competing disproportionation yielding triethylbenzene, represented by the red bonds in all structures. **c** Structures of the OSDA mimic and of the diaryl intermediates for transalkylation ($I_{trans}$) and disproportionation ($I_{disp}$).

selective structures contain 1D channel systems (orange points) or too small 8-ring windows (blue points) and therefore are not included in our selected set of candidates. However, to confirm or refute the expected behavior of structures with 1D channels systems MOR, which has been tested experimentally and used industrially, was included in the dataset.

To improve the accuracy of these data, the neutral diaryl intermediates for transalkylation ($I_{trans}$) and disproportionation ($I_{disp}$) were optimized inside the twelve selected zeolite structures listed in Table 1 using periodic DFT calculations. In these simulations, which include multiple interactions other than dispersion, the zeolite pores were filled with the maximum number of molecules possible, given in column n in Table 1. The optimized geometries of the $I_{trans}$ intermediates in the twelve zeolites considered are shown in Fig. 2. Despite the differences in methodology and model, the FF and DFT values follow the same relationship in the plot of BE $I_{disp}$ versus BE $I_{trans}$ in

Supplementary Fig. 2c, thus confirming the validity of the trend proposed by the computationally cheaper and faster FF calculations.

The DFT binding energies per mole of diaryl intermediate are in all cases negative and close to or larger than −100 kJ/mol, indicating that all the zeolite structures selected are well-suited to stabilize this type of intermediate and therefore should in principle favor the diaryl-mediated pathway of the mechanism. The BE ratio $I_{disp}/I_{trans}$ provides quantitative information about the potential selectivity of each zeolite structure. Values larger than 1 indicate a better stabilization of $I_{disp}$ intermediate as compared to $I_{trans}$, and therefore a larger selectivity to the undesired TEB. It is evident from Table 1 that zeolites containing large channels like ITT (18x10x10) or UTL (14×12) clearly favor the bulkier intermediate involved in the disproportionation, followed by BEC, USI, IWR, and SEW with 2D or 3D channels systems composed by 12-ring and 10-ring channels. In contrast, zeolites with 1D pores like MOR disfavor the formation of the $I_{disp}$ intermediate as compared to

**Table 1 | DFT calculated BE of neutral diaryl intermediates for transalkylation ($I_{trans}$) and disproportionation ($I_{disp}$) in zeolites with different microporous structure**

|  | Channels system | n | $I_{trans}$ (kJ/mol) | $I_{disp}$ (kJ/mol) | $I_{disp}$ /$I_{trans}$ |
|---|---|---|---|---|---|
| BEA | 12×12×12 | 3 | −148 | −129 | 0.87 |
| BEC | 12×12×12 | 1 | −123 | −151 | 1.22 |
| BOG | 12×10×10 | 4 | −140 | −140 | 1.00 |
| CON | 12×10×10 | 2 | −132 | −149 | 1.14 |
| FAU | 12×12×12 | 7 | −104 | −106 | 1.02 |
| ITT | 18×10×10 | 4 | −110 | −155 | 1.41 |
| IWR | 12×10×10 | 1 | −145 | −174 | 1.20 |
| IWV | 12×12 | 2 | −102 | −90 | 0.88 |
| MOR | 12×8 | 2 | −141 | −104 | 0.74 |
| SEW | 12×10 | 2 | −160 | −188 | 1.18 |
| USI | 12×10 | 2 | −128 | −157 | 1.22 |
| UTL | 14×12 | 2 | −105 | −137 | 1.31 |

$I_{trans}$. Notice that the formation of the bulkier $I_{disp}$ intermediate is not hindered in any zeolite as indicated by the large negative values of BE, and therefore the classical concept of shape-selectivity[26] cannot be applied in this case. According to these data IWV and BEA, with 2D and 3D channels systems of 12-rings, should be intrinsically selective to EB.

## DFT study of the reaction mechanism

The next step to find the optimum zeolite was to evaluate the kinetics of the two competing reactions by means of a DFT study of the reaction mechanism. It has been established in previous work dealing with the transalkylation reaction[23,24] that the proton transfer between the two aromatic rings of the cationic diaryl intermediate $I1^+$ to form $I4^+$ can follow at least the four different pathways depicted in Supplementary Figs. 3 and 4. Taking into account the relative stability of the ortho-, meta- and para-isomers of neutral, cationic, and diaryl intermediates involving DEB (see Supplementary Table 2), the most stable isomer p-DEB$^+$ cation was assumed to react with benzene in the transalkylation reaction, and with ortho-, meta- and para-DEB in the disproportionation reaction (see Supplementary Tables 3 and 4). All calculated energy profiles for the four pathways considered are plotted in Supplementary Fig. 5, and only the lowest energy routes for transalkylation and disproportionation are plotted in Fig. 3 and discussed in the main text.

The proton shifts between carbon atoms belonging to the same aromatic ring occurring through transition states TS2 and TS4 are fast, and therefore the highest activation energy barrier in each pathway always corresponds to the transfer of the H$^+$ from one aromatic ring of the cationic diaryl intermediate to the other one. The most favorable route for transalkylation (black lines in Fig. 3) is the three-step process in which the inter-ring H$^+$ transfer takes place via a six-membered transition state, TS3, with an activation energy of 56 kJ/mol. The direct H$^+$ transfer through a four-membered transition state, TS1, with an activation barrier of 76 kJ/mol, and the routes via five-membered transition states TS5 and TS6, with calculated barriers of 72 and 91 kJ/mol, respectively, are less favored. On the other hand, higher activation energies ranging from 80 to 116 kJ/mol are obtained for all disproportionation pathways, due in part to better stabilization of the positive charge in more substituted C atoms of the aromatic rings, as for instance in $I2^+$ (see schematic charge location in Fig. 3). These results indicate that even in the absence of any zeolite confinement effect, disproportionation is intrinsically more energetically demanding than transalkylation.

To analyze the influence of the zeolite framework on the kinetics of the transalkylation and disproportionation reactions, the four pathways described above were investigated in pure silica models of

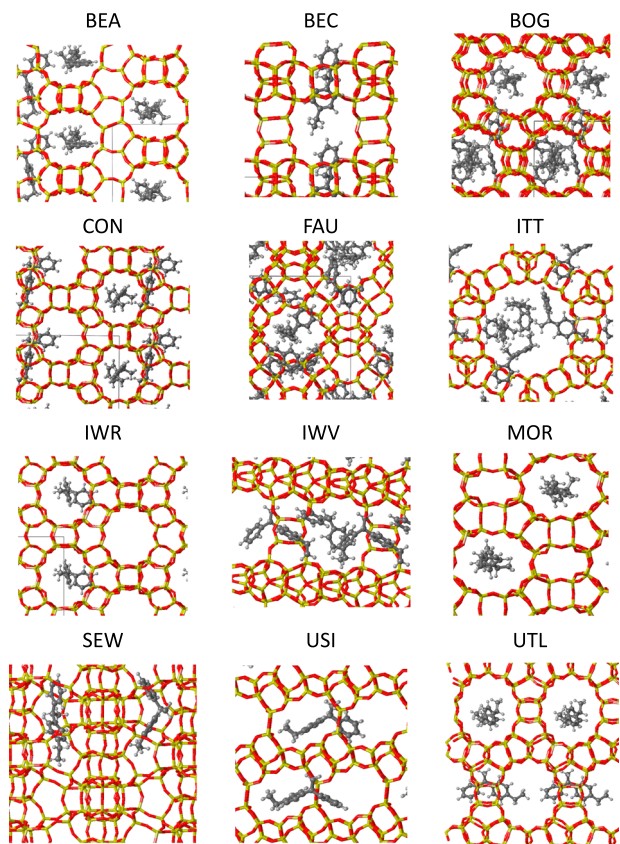

BEA BEC BOG
CON FAU ITT
IWR IWV MOR
SEW USI UTL

**Fig. 2 | Zeolite structures selected from the initial FF computational screening.** DFT optimized structures of the diaryl intermediate for transalkylation $I_{trans}$ in the twelve zeolites investigated. Si, O, C, and H atoms are represented as yellow, red, gray, and white balls, respectively.

the promising BEC, BOG, IWR, IWV, and UTL zeolites using periodic DFT calculations. This selection includes structures with 2D and 3D channels systems containing 10-, 12-, and 14-ring channels, all of them with good BE for the neutral transalkylation $I_{trans}$ intermediate. MOR, with a 1D channel system where the diaryl intermediates do not match too well, was included for comparison. To analyze in detail the influence of the channel size on the kinetics of the reaction, the mechanistic study in UTL was performed in two different locations, in the large 14-ring channel (UTL(cha)) and at the intersection between the 12-ring and 14-ring channels (UTL(int)). In order to discard any deviations in the calculated energies associated to interactions between net positive charges in periodically repeated cells, a comparison with previous[24] and new data obtained with Al-containing neutral models is presented in Supplementary Tables 5 and 6 and Supplementary Figs. 6–9. The comparison confirms the accuracy of the approach followed.

The relative stability of all minima and transition states are summarized in Supplementary Table 7, and the corresponding energy profiles are plotted in Fig. 4a. The reaction paths within the zeolite channels are equivalent to those described in gas phase and only small differences in relative energies of intermediates and in activation barriers are found due to the steric restrictions imposed by the zeolite frameworks. To facilitate comparison, the activation energies for the six elementary steps investigated are summarized in Table 2. As in the gas phase study, the barriers for the intra-ring H$^+$ shifts, Ea2 and Ea4, are between 20 and 50 kJ/mol in all zeolites, confirming that the H$^+$ transfer between the two aromatic rings is the rate-determining step also in confined spaces. Applying the energy-span model to compare different zeolite structures and preferred pathways, the values of Ea1, Ea3, Ea5, and Ea6, representative of each route, are plotted together in Fig. 4b.

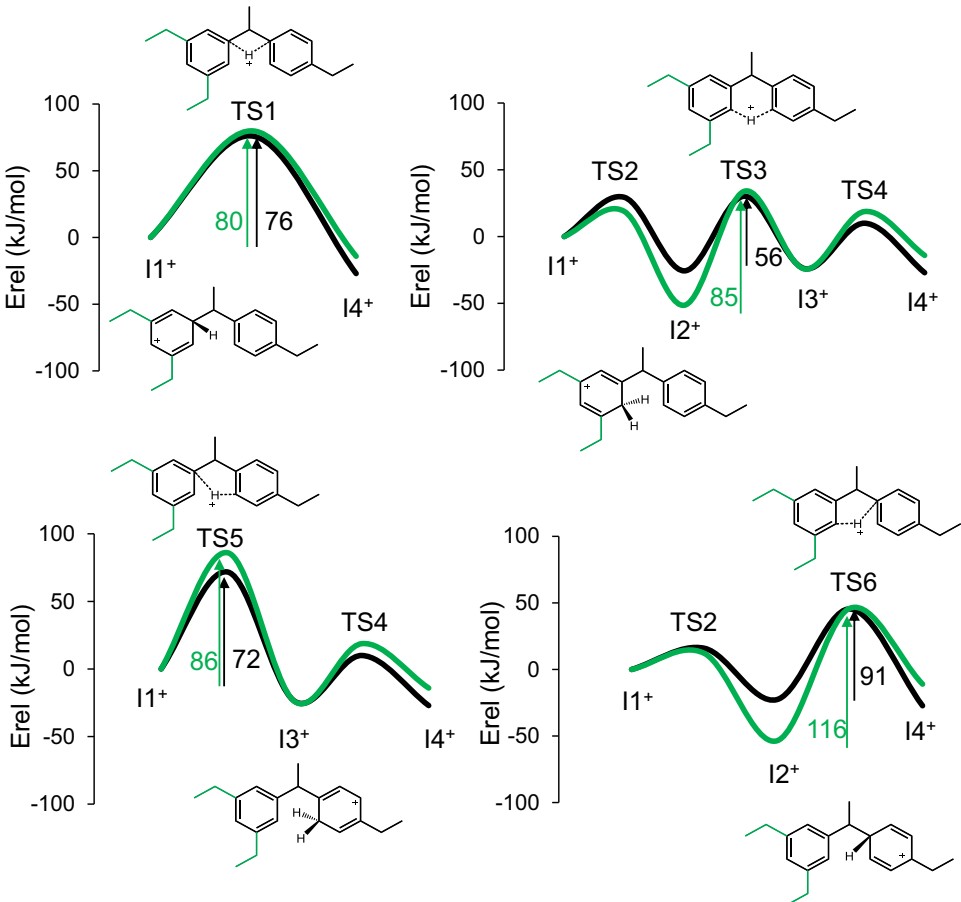

**Fig. 3 | DFT results for the diaryl-mediated pathway.** Calculated energy profiles for transalkylation (black) and disproportionation (green) pathways in gas phase and schematic representation of transition states and intermediates. The highest activation energy barrier in each pathway is given beside the arrows (in kJ/mol).

The lowest energy pathway for transalkylation is in all cases the three-step process taking place through TS3, with activation energies Ea3 ranging from 44 kJ/mol in BEC to 78 kJ/mol in MOR. In most zeolite structures considered, the second preferred pathway is the direct H⁺ transfer through TS1 with the calculated Ea1 values being ~10 kJ/mol larger than the corresponding Ea3 values (Table 2). Two clear exceptions are BEC and MOR, with activation energies for this step close to 100 kJ/mol. Finally, the activation barriers for the routes involving five-membered transition states, Ea5 and Ea6, are larger than 80 kJ/mol, with the only exception of Ea5 in IWV (60 kJ/mol) and UTL(cha) (68 kJ/mol).

From a geometrical point of view, the most demanding step is the direct H⁺ transfer through a four-member ring, as depicted in Fig. 5. In the gas phase, that is, in the absence of any steric constraint, the α angle closes from 108° in I1⁺ to 101° in TS1, which induces a closing of the β angle from 128° to 98°, while the γ and δ angles around 180° reflect the planarity of the two aromatic rings. If these changes are not possible within the zeolite voids, further deformations occur with the associated energy penalty, as exemplified in Fig. 5 (see all angles in Supplementary Table 8). The optimized structure of TS1 in IWV is the same as in the gas phase, and only small variations of ~5° in α and β are observed in IWR. In contrast, the diaryl structure must flatten in the unidirectional 12-ring channels of MOR, resulting in an increase of the β angle to 113° and a decrease of the γ and δ angles to 170° and 172°, respectively. A similar deviation from the optimum gas phase geometry of TS1 is forced inside the microporous structure of BEC, which despite being a 3D system of 12-ring channels does not form any large void at the channels intersection, thus explaining the significantly

higher Ea1 values calculated in these two structures. A different situation is found for TS3, where the six-member ring involved in the H⁺ transfer accommodates better the deformations required to fit within the zeolites' voids (see Supplementary Table 8). The largest deviations from the gas phase values of α and β for TS3 are found in MOR (3° and 6°) and BOG (2° and 5°), the two zeolites exhibiting the highest Ea3 barriers in Table 2.

In order to obtain a unique quantitative parameter able to characterize the kinetic behavior of each zeolite structure in the transalkylation reaction, a Boltzmann weighted average of the activation energies for each of the four inter-ring H⁺ transfer steps considered was calculated. At 240 °C, which is the experimental reaction temperature, only the lowest energy pathway through TS3 contributes significantly to the average activation energy. Taking Ea3 activation energy as predictor of zeolite activity, the following order is obtained for the transalkylation reaction: BEC > UTL(int) > IWV >> UTL(cha) > IWR > BOG > MOR. Notice that this is not the same order of catalytic performance provided by the BE of the I_trans intermediate: IWR > MOR > BOG > BEC > UTL > IWV. This discrepancy in the predicted catalytic performance of similar zeolites suggests that the stability of the main intermediate of the reaction mechanism is a good parameter for an initial screening of potential catalysts, but perhaps is not specific enough to describe accurately the outcome of such a complex reaction network.

To check whether the selectivity to EB and TEB is well described by the stabilization of the corresponding intermediate (I_trans and I_disp) or also changes when considering more accurate kinetic data, the four pathways for the disproportionation of DEB were calculated in the a

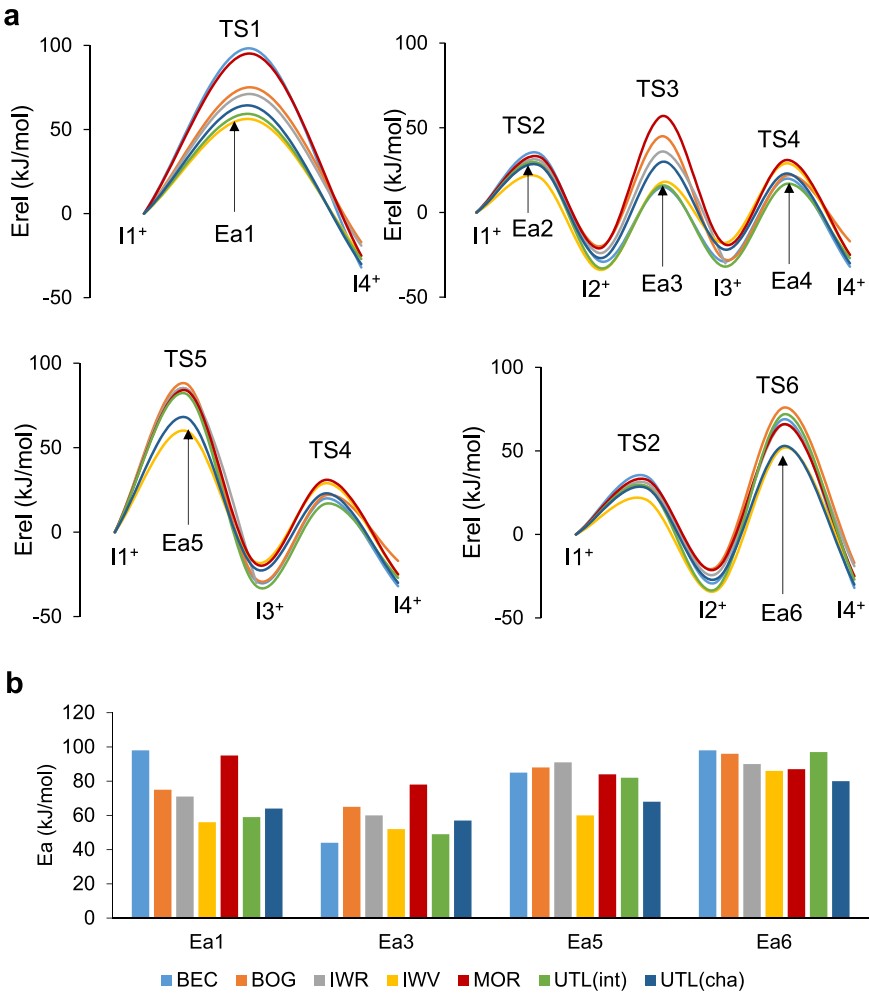

**Fig. 4 | DFT profiles for the transalkylation reaction. a** Calculated energy profiles and **b** comparison of the activation energy barriers for transalkylation pathways in BEC, BOG, IWR, IWV, MOR and UTL zeolite structures.

**Table 2 | Calculated activation barriers (in kJ/mol) for all the elementary steps of the transalkylation and disproportionation mechanisms in different zeolite structures**

|  | Channels system | Ea1 | Ea2 | Ea3 | Ea4 | Ea5 | Ea6 |
|---|---|---|---|---|---|---|---|
| transalkylation |  |  |  |  |  |  |  |
| BEC | 12 × 12 × 12 | 98 | 35 | 44 | 48 | 85 | 98 |
| BOG | 12 × 10 × 10 | 75 | 29 | 65 | 50 | 88 | 96 |
| IWR | 12 × 10 × 10 | 71 | 31 | 60 | 44 | 91 | 90 |
| IWV | 12 × 12 | 56 | 21 | 52 | 47 | 60 | 86 |
| MOR | 12 × 8 | 95 | 33 | 78 | 50 | 84 | 87 |
| UTL(int) | 14 × 12 | 59 | 29 | 49 | 49 | 82 | 97 |
| UTL(cha) | 14 × 12 | 64 | 28 | 57 | 45 | 68 | 80 |
| disproportionation |  |  |  |  |  |  |  |
| BEC | 12 × 12 × 12 | 113 | 30 | 56 | 55 | 78 | 118 |
| IWV | 12 × 12 | 72 | 30 | 75 | 59 | 67 | 104 |
| UTL(int) | 14 × 12 | 90 | 29 | 69 | 58 | 71 | 98 |
| UTL(cha) | 14 × 12 | 80 | 22 | 73 | 46 | 82 | 97 |

priori most selective IWV, less selective UTL and intermediate BEC zeolite structures considered in the mechanistic study. The trends observed in gas phase are reproduced inside the zeolite micropores (Table 2, Fig. 6, and Supplementary Table 3). All activation barriers for disproportionation are systematically higher than for transalkylation, and the localization of the positive charge in a more substituted carbon atom of the aromatic ring produces an additional stabilization of the $I2^+$ intermediate for disproportionation as already observed in the gas phase study. In an attempt to quantify this effect, the difference between the lowest activation energies for disproportionation and transalkylation were computed for BEC, IWV, UTL(int) and UTL(cha). The calculated values are quite similar, 12, 15, 20 and 16 kJ/mol, respectively, suggesting that the steric impediments in each zeolite affect similarly all intermediates and transition states. Therefore, the selectivity to TEB will probably be well described by the relative stabilization of the main $I_{trans}$ and $I_{disp}$ intermediates inside the zeolite pores.

Finally, the complete mechanism for the transalkylation reaction following the alkyl-transfer route was calculated in BOG and UTL and compared with results published previously for MOR and IWV[24] in order to confirm that this pathway is not competitive in the zeolite structures selected to stabilize diaryl intermediates. The values in Supplementary Table 9 and the profiles depicted in Supplementary Fig. 10 confirm that the ethyl transfer between the zeolite framework and the organic fragment is the most energetically demanding step, with TS8 and TS9 being between 135 and 160 kJ/mol higher in energy than the initial reactants. Such high activation energies indicate the contribution of the alkyl-transfer pathway to DEB conversion should be low or negligible in the selected zeolite structures.

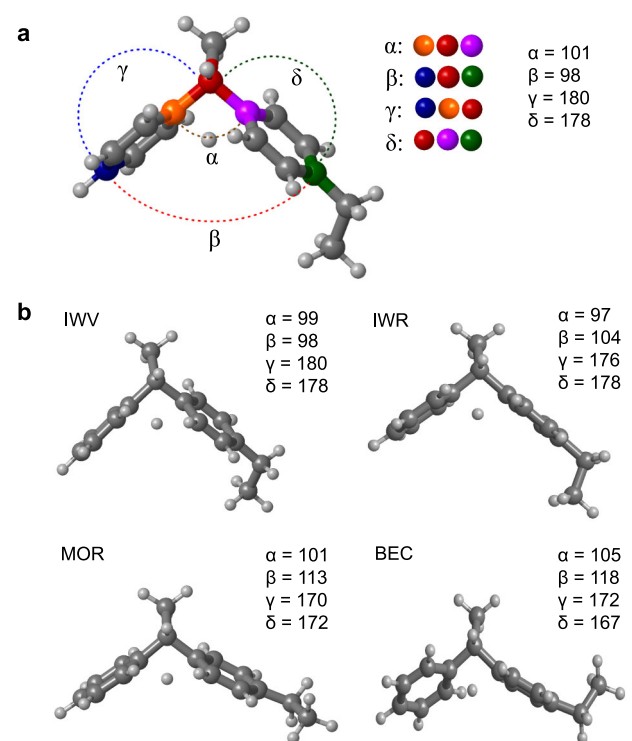

**Fig. 5 | Geometrical distortion of transition states in different zeolite channels.**
**a** Definition of the α, β, γ, and δ angles and optimized values corresponding to TS1 in the gas phase. The different colors are used to define the α, β, γ, and δ angles. **b** Optimized geometries of TS1 in IWV, IWR, MOR, and BEC zeolites with the corresponding values of the α, β, γ, and δ angles.

## Synthesis and catalytic test of proposed zeolites

To confirm or discard the trends proposed by the computational screening, the following zeolites from the set in Table 1 were synthesized and their catalytic activity tested: extra-large pore ITQ-33 (ITT) and ITQ-15 (UTL), large-pore containing super-cages USY (FAU) large-pore with 3D channels ITQ-17 (BEC), large-pore with 2D channels ITQ-27 (IWV), intersecting large and medium-pore ITQ-47 (BOG) and ITQ-24 (IWR), and finally 1D large-pore mordenite (MOR).

PXRD patterns of all samples showed characteristic diffraction peaks of the corresponding structures with high crystallinity and without impurities (see Supplementary Fig. 11), and $N_2$ adsorption measurements also reveal the expected textural properties for each material (see Supplementary Table 10). According to FESEM images, the entire synthesized zeolites have particle sizes below 1 µm, being most of them within 200–500 nm (Supplementary Fig. 12). Tetrahedrally-coordinated Al species are preferably found in the different zeolites according to $^{27}$Al MAS NMR spectra (Supplementary Fig. 13), and the pyridine adsorption/desorption IR-spectra revealed that the amount of Brønsted acid sites is proportional to the Al content of the samples (Supplementary Table 10).

The catalytic tests were performed under gas phase conditions to avoid surface saturation that will happen under liquid phase conditions. The level of DEB conversion was adjusted by varying contact time (w/F, Supplementary Table 11) to below 20% in order to obtain initial reaction rates of transalkylation ($r_{trans}$ in Table 3). Under such conditions, the main product is in all cases ethylbenzene, with selectivity ranging from 75 to 95%, followed by triethylbenzene and ethene as the main byproducts. Heavy products, mainly highly alkylated aromatics were also observed in low amounts.

As expected, zeolites containing large voids within their structure, like ITQ-33 (ITT, with 18-ring channels of 12.3 Å diameter)[27] and ITQ-15 (UTL, with 14-ring channels of 9.5 × 7.1 Å) produce large amounts of

TEB, 20.8%, and 12.0%, respectively. Significant amounts of TEB, ~8%, are also formed in BEC with a 3D system of 12-ring channels of 7.5 × 6.3 Å, but not in FAU, with a supercage of 11.4 Å diameter. Moving to zeolites containing 12-ring channels of 7.0 Å or less, such as ITQ-47 (BOG, 7.0 × 7.0 Å), MOR (7.0 × 6.5 Å), IWV (6.9 × 6.2 Å) and ITQ-24 (IWR, 6.8 × 5.8 Å), results in a clear decrease of TEB formation probably due to steric problems to form the bulky diaryl intermediates involved in DEB disproportionation. Indeed, a fairly good correlation between the selectivity to TEB and EB determined experimentally (TEB/EB ratio calculated from data in Table 3) and the theoretical BE ratios $I_{disp}/I_{trans}$ given in Table 1 is observed in Fig. 7a). Only two structures are out of this trend. MOR produces more TEB than predicted by the BE ratios of the diaryl intermediates because the aromatics transformations in this unidirectional zeolite proceed in part through the alkyl-transfer pathway, as confirmed by the observation of 4.6% ethene (Table 3). In contrast, less TEB than expected from the theoretical BE ratios $I_{disp}/I_{trans}$ is experimentally detected in IWR. The reason is the tight fitting of the diaryl intermediate for disproportionation $I_{disp}$ at the intersection between the 12-ring and the two 10-ring channels. At this specific position, the two ethyl groups in one of the aromatic rings fit perfectly into the two narrow 10-ring channels of IWR leading to an excellent stabilization of $I_{disp}$ but hindering the diffusion of TEB out of the crystal.

The rate of transalkylation ($r_{trans}$) was measured as moles of EB formed per hour normalized by the moles of Brønsted acid sites present in each sample, and no clear trend with either the dimensionality or size of the channels system is observed at first sight. The highest reaction rate corresponds to ITQ-27 (IWV, 12 × 12), followed by ITQ-15 (UTL) with larger 14 × 12-ring pores. In zeolites with wider pores like ITQ-33 (ITT, 18 × 10 × 10) or containing large cavities within their structure, like FAU, the lower transalkylation rate could be associated to a worse fitting of the intermediates and transition states within the large voids, and therefore to their weaker stabilization by confinement. On the other hand, ITQ-24 (IWR) with a narrower three-dimensional 12 × 10 × 10 channel system exhibits a transalkylation rate similar to that of ITQ-17 (BEC), with wider 12 × 12 × 12 channels but without any large cavity at the channels' intersection. In contrast, the catalytic activity of ITQ-47 (BOG) also with a narrow 12 × 10 × 10 system of channels is much lower than that of IWR and comparable to that of MOR with uni-directional channels. Interestingly, the selectivity to ethene in BOG is 4.8%, indicating some contribution of the alkyl-mediated pathway. On the other hand, there seems to be a rough trend in the plot of the experimental reaction rates for transalkylation ($r_{trans}$) and the calculated BE for $I_{trans,}$ with higher reaction rates in the zeolite structures that stabilize less the $I_{trans}$ intermediates (Fig. 7b) but the trend is not consistent along the dataset and the correlation is not good.

Apparent activation energies (Ea) for each catalyst were calculated from the initial reaction rates ($r_{trans}$ measured at four different temperatures using the Arrhenius equation. The reaction rate constants ($k$) determined at each temperature are listed in Supplementary Table 12, the plots of lnk versus 1/T are shown in Supplementary Fig. 14, and the calculated Ea values are summarized in Table 3. To connect the experimental and computational information, the apparent activation energies (Ea) and the reaction rates for transalkylation ($r_{trans}$) were plotted against the DFT calculated Ea3 activation energies. The good correlation observed between the experimental and theoretical activation energies (Fig. 7c) indicates that in these zeolite structures the DEB-benzene transalkylation reaction occurs preferentially through formation and transformation of diaryl intermediates. On the other hand, the experimental reaction rates for transalkylation decrease as the calculated Ea3 activation energies increase, with only one structure, BEC, out of this trend (Fig. 7d). The reason is that the synthesis of some of these large-pore zeolites is only possible if Ge is incorporated in the framework. In particular, the low Si/Ge ratio determined for ITQ-

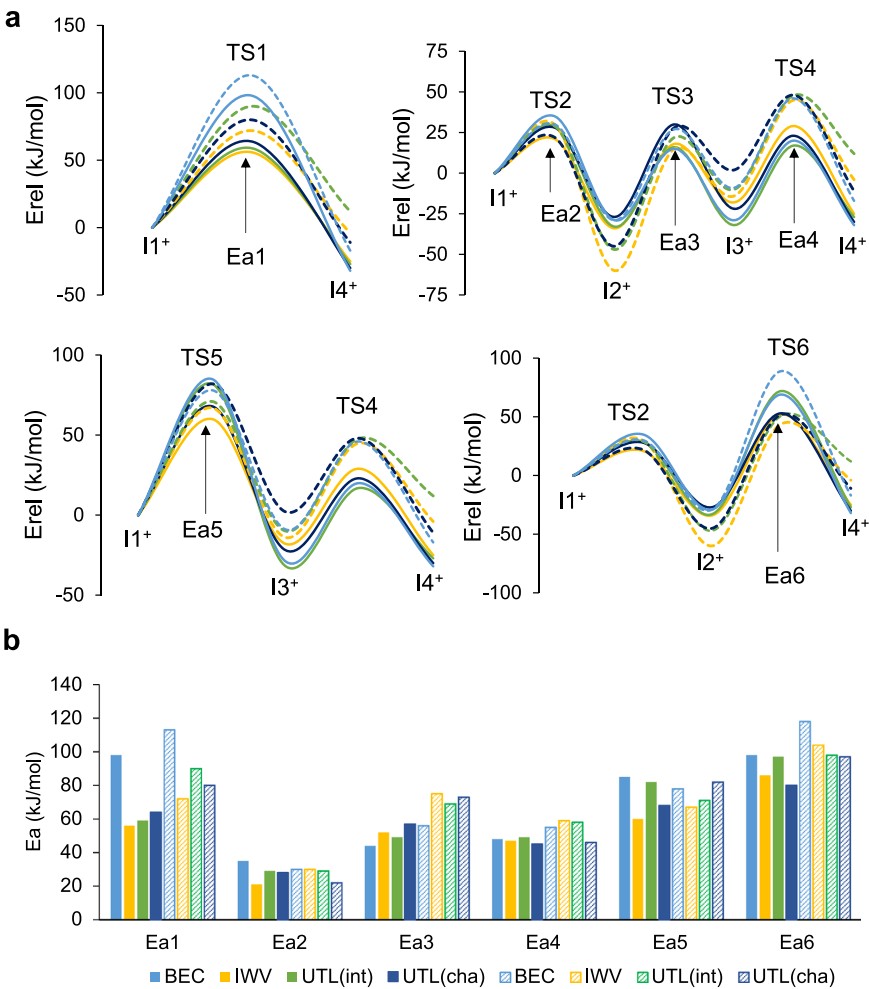

**Fig. 6 | DFT profiles for the disproportionation reaction. a** Calculated energy profiles and **b** comparison of the activation energy barriers for transalkylation (full lines and columns) and disproportionation (dashed lines and columns) in BEC, IWV and UTL zeolites.

**Table 3 | Results of catalytic test in diethylbenzene-benzene transalkylation at 240 °C**

| zeolite | IZA code | DEB conv. (%) | $r_{trans}$ (mol$_{EB}$/mol$_{acid}$h) | EB (%) | TEB (%) | C$_2$H$_4$ (%) | HP (%) | Ea$^a$ (kJ/mol) |
|---|---|---|---|---|---|---|---|---|
| ITQ-33 | ITT | 11.9 | 717 | 76.4 | 20.8 | 0.6 | 2.1 | 75.2 |
| ITQ-15 | UTL | 17.7 | 1599 | 85.7 | 12.0 | 0 | 2.3 | 56.1 |
| USY | FAU | 14.7 | 1075 | 90.1 | 2.7 | 2.7 | 4.5 | 66.5 |
| ITQ-17 | BEC | 20.8 | 628 | 88.8 | 8.3 | 0.5 | 2.3 | 59.5 |
| ITQ-27 | IWV | 14.9 | 1926 | 94.1 | 0.5 | 2.3 | 3.0 | 58.3 |
| ITQ-24 | IWR | 15.2 | 676 | 95.1 | 2.5 | 0.4 | 2.0 | 69.4 |
| ITQ-47 | BOG | 14.4 | 349 | 90.4 | 4.4 | 4.8 | 0.4 | 66.2 |
| mordenite | MOR | 9.6 | 279 | 88.6 | 2.5 | 4.6 | 4.2 | 74.2 |

$^a$Calculated using the Arrhenius equation.

17 (BEC) sample (Supplementary Table 10) could result in a low framework stability that modifies the catalytic activity, breaking the correlation between reaction rates and DFT calculated activation energies.

## Discussion

In this work, we have presented a computational methodology for the "in silico" selection of the most adequate existing zeolites as catalysts for a given reaction. The methodology combines a fast screening based on force fields of all zeolite structures able to stabilize the key intermediates of potentially competing reactions with a more

demanding periodic DFT study of the reaction mechanisms on the most promising catalyst candidates. It has been applied exemplarily to competing DEB transalkylation and disproportionation. Then, the large pore zeolites with different microporous structures of channels and cavities proposed by the computational study have been synthesized, characterized, and their catalytic activity tested. Comparison of the theoretical and experimental results validates the proposed methodology and leads to the following conclusions.

The experimentally observed selectivity to TEB and EB in each zeolite correlates with the relative stability of the diaryl intermediates involved in transalkylation (I$_{trans}$) and disproportionation (I$_{disp}$)

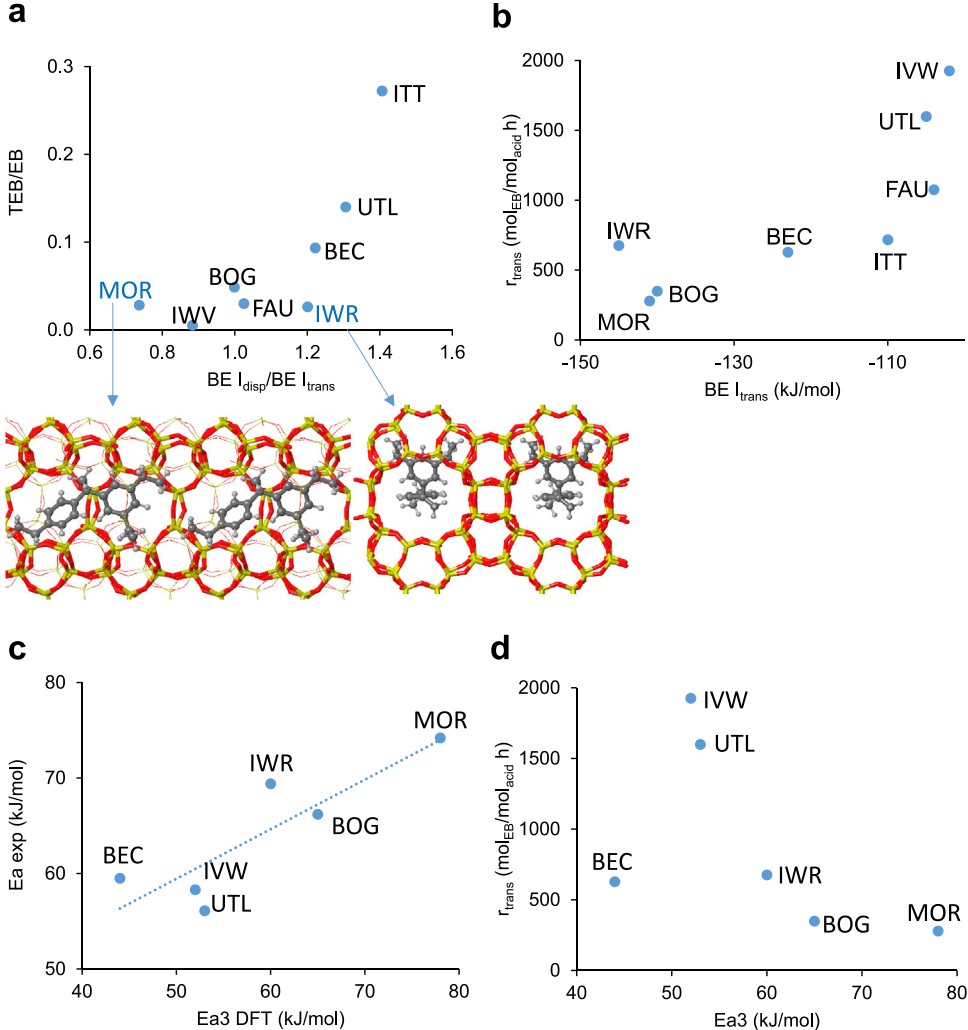

**Fig. 7 | Connection between computational data and experimental results.**
**a** Correlation between the experimental TEB/EB selectivity and the computational
BE $I_{disp}$/$_{trans}$ ratio. The optimized geometry of the diaryl $I_{disp}$ intermediate inside
the channels of MOR and IWR structures is shown. **b** Plot of experimental reaction
rates for transalkylation ($r_{trans}$) versus DFT calculated BE of the diaryl $I_{trans}$ inter-
mediate. **c** Correlation between experimental and DFT calculated activation ener-
gies. **d** Correlation between experimental reaction rates for transalkylation ($r_{trans}$)
and DFT calculated activation energies.

determined by their calculated BE. It goes beyond the classical shape-
selectivity or size-exclusion concept according to which some reac-
tions are prevented because the intermediates or transition states
involved require more space than available, and it is explained instead
by a better fitting of one of the two slightly different diaryl inter-
mediates within the zeolite microporous structure, even if both of
them can be accommodated.

The catalytic activity of each zeolite, experimentally given by the
rate of transalkylation ($r_{trans}$), correlates with the calculated activation
energy for the rate-determining step of the pathway that contributes
most at the reaction temperature. Therefore, an accurate description
of the catalytic activity requires a mechanistic study and the calcula-
tion of the kinetics of the reaction.

The transalkylation of DEB with benzene in large pore zeolites
containing 2D and 3D systems of 12-ring channels occurs preferentially
following the diaryl-mediated pathway, while some contribution of the
alkyl-transfer route is expected in uni-directional 12-ring channels and
in smaller 10-ring channels. The internal architecture of the IWV zeolite
is the best suited for this reaction.

In summary, a new tool for the "in silico" study of competing
reactions catalyzed by zeolites that allows the selection of the most
adequate catalyst for a given reaction is presented. This methodol-
ogy can recognize and quantify subtle differences in the

stabilization of intermediates and transition states within similar
microporous voids, thus approaching the level of molecular recog-
nition of enzymes. This tool can be safely applied to propose the
most adequate zeolite structures, either existing or hypothetical, for
target reactions, leading to a significant reduction of expensive
experimental work.

## Methods
### Force field calculations
Force field calculations were performed using the General Utility Lat-
tice Program (GULP), version 5.1.1[28,29], through the GULPy package[30].
Initial zeolite structures were downloaded from the International
Zeolite Association (IZA) database[27] and pre-optimized using the
Sanders-Leslie-Catlow (SLC) parametrization[31]. SMILES strings for
OSDAs were extracted from the literature[32]. Conformers for OSDAs
were generated using RDKit[33] with the MMFF94 force field[34,35] after
explicit enumeration of all stereoisomers for each molecule.

Generation of OSDA-zeolite and intermediate-zeolite poses was
performed using the Voronoi and Monte Carlo docking algorithms as
implemented in the VOID package[36]. At most 5 different conformers
for each OSDA and intermediate are used as input guest geometries for
VOID. For certain zeolites with lattice parameters smaller than 10 Å in
one direction, unit cells and supercells elongated in the direction of

the short parameter by factors of 2 and 3 were used as inputs for the docking package. The Voronoi docking algorithm used a threshold of 3 Å for Voronoi node clustering and a threshold fitness function with a minimum distance of 1.25 Å. At first, docking is performed with the batch Voronoi docker parallelized over 20 images. In the case of OSDAs, loading is increased until no more conformers can be added to the pose without clashing, and repeated if the procedure does not yield any poses. In the case of reaction intermediates, a single molecule is placed inside the zeolite using the same docking procedure. In this case, the loading is not increased, but different configurations of the molecule placement inside the framework are explored. The Dreiding force field[37] was used to model interactions between the zeolite and the OSDA. Structural optimizations of poses were performed at constant volume using the conjugate gradient and rational function optimization algorithms, switching to the latter when the norm of the gradient dropped below 0.10 eV/Å. The frozen pose method[38] was then used to compute binding energies. Poses with binding energy larger than zero were considered to have unfavorable host-guest interactions.

## DFT calculations

All calculations in the gas-phase mechanistic study are based on density functional theory (DFT) and were carried out using the M062X functional[39] and the 6–31 g(d,p) basis set[40], as implemented in the Gaussian09 software[41]. The positions of all C and H atoms in the cationic intermediates and transition states were fully optimized without restrictions, and all stationary points were characterized by means of harmonic frequency calculations.

Periodic density functional theory (DFT) calculations were performed using the Perdew-Burke-Ernzerhof (PBE) exchange-correlation functional within the generalized gradient approach (GGA)[42,43], as implemented in the Vienna Abinitio Simulation Package (VASP) code[44]. The valence density was expanded in a plane wave basis set with a kinetic energy cutoff of 600 eV, and the effect of the core electrons in the valence density was taken into account by means of the projected augmented wave (PAW) formalism[45]. Integration in the reciprocal space was carried out at the Γ k-point of the Brillouin zone. Dispersion corrections to the energies were evaluated using the D3 Grimme's method[46,47]. Electronic energies were converged to $10^{-6}$ eV and geometries were optimized until forces on atoms were smaller than 0.01 eV/A. Transition states were obtained using the DIMER and NEB algorithms[48–51]. During geometry optimizations, the positions of all atoms in the system were allowed to relax without restrictions. The symmetry and unit cell parameters of the zeolite models employed in the periodic DFT study are summarized in Supplementary Table 13.

## Synthesis of zeolites

**FAU.** The commercially-available CBV720 sample from Zeolyst was employed.

**MOR.** A total of 13.4 g of tetraethylammonium bromide (TEABr, Sigma-Aldrich), 8.76 g sodium aluminate (Carlo Erba), and 10.2 g NaOH were dissolved in 344.3 g deionized water. To this solution, 84.8 g fumed silica was added and kept under mechanic stirring until a viscous white gel is obtained. Then, this gel was transferred to a Teflon-lined autoclave and heated under rotation at 150 °C for 7 days. The solid product was then separated by filtration, washed with abundant water and then dried in air at 60 °C.

In order to prepare the proton form of this zeolite for catalysis, the zeolite was first calcined in a muffle at 550 °C for 6 h. The solid was mixed with 1 mol/L ammonium nitrate solution at a liquid/solid mass ratio of 50 and maintained at room T for 16 h. Finally, the solid was separated by filtration and dried at 60 °C overnight. Finally, the material was calcined at 500 °C for 3 h in a muffle.

**ITQ-15 (UTL).** A typical synthesis gel was first prepared by dissolving 0.418 g of $GeO_2$ in 13.38 g of the solution of the ammonium salt derivate from L-Proline in hydroxide form with a concentration of 0.9 mol $OH^-/kg$[52]. Then, 3.0 g of Ludox (40 wt.% of colloidal silica suspension in $H_2O$) was added, and the mixture was maintained under stirring until achieving the following gel molar composition: $0.833\ SiO_2$: $0.167\ GeO_2$: 0.5 OSDA(OH): 10 $H_2O$, where OSDA(OH) is the ammonium salt derivate from L-Proline in the hydroxide form. Additionally, 0.06 g of ITQ-15 crystals was added as seeds. The synthesis gel was autoclaved at 175 °C for 4 days under stirring conditions. The resultant solids were recovered by filtration, washed with distilled water, dried at 100 °C for 12 h, and characterized. The incorporation of aluminum was carried out by a post-synthesis treatment[52]. The as-prepared ITQ-15 was calcined at 580 °C for 3 h. Then, 0.5 g of calcined ITQ-15 zeolite was mixed with 7.5 ml of an $Al(NO_3)_3$ solution (5wt. % in $H_2O$) in an autoclave, which was placed at 150 °C for 18 h. The Al-containing ITQ-15 was recovered by filtration, washed with distilled water, dried at 100 °C for 12 h, and characterized.

**ITQ-24 (IWR).** The synthesis of ITQ-24 zeolite was carried out using hexamethonium dihydroxide as OSDA. The gel composition was: $SiO_2$: $0.04\ B_2O_3$: 0.25 $OSDA(OH)_2$: 0.5 $NH_4F$: 3 $H_2O$, where $OSDA(OH)_2$ is hexamethonium dihydroxide. In a typical synthesis, 0.307 g of boric acid ($H_3BO_3$) was dissolved in 0.94 g of a hexamethonium hydroxide solution (18.84% wt.). Then, 0.458 g of Ludox AS-40 and 0.558 g of $NH_4F$ solution (10% wt.) were added. The mixture was stirred to evaporate water until the desired gel composition was reached. Finally, 0.045 g of ITQ-24 zeolite was added as seed and the gel was heated for 14 days at 175 °C.

The incorporation of aluminum was done by a post-synthesis treatment[52]. The as-prepared ITQ-24 was calcined at 580 °C for 3 h. Then, 0.5 g of calcined ITQ-24 zeolite was mixed with 7.5 ml of an $Al(NO_3)_3$ solution (2wt. % in $H_2O$) in an autoclave, which was placed at 150 °C for 18 h. The Al-containing ITQ-24 was recovered by filtration, washed with distilled water, dried at 100 °C for 12 h and characterized.

**ITQ-47 (BOG).** This zeolite was synthesized following a procedure reported by Simancas et al. using P1-phosphazene as OSDA[53]. The composition of the synthesis gel was: 0.833 TEOS: 0.167 $GeO_2$: 0.0370 $H_3BO_3$: 0.40 P1-phosphazene: 10 $H_2O$, where TEOS is tetraethylorthosilicate (TEOS, Sigma-Aldrich, 98%wt). This gel was transferred to Teflon lined stainless-steel autoclaves and heated to 150 °C under tumbling for 25 days. The resulting solid was filtered and washed exhaustively with distilled water. The resulting solid was dried at 100 °C overnight and, afterwards, calcined in air at 550 °C. The calcined material was treated with an 8 wt.% $Al(NO_3)_3$ aqueous solution, and the resulting mixture was transferred to a Teflon lined stainless-steel autoclave and kept heated to 140 °C under tumbling for 3 days. The resulting solid was filtered and exhaustively washed with distilled water, dried at 100 °C for 12 h, and, finally, calcined in air at 550 °C.

**ITQ-17 (BEC).** The Al-containing form of the polymorph C of Beta zeolite (Al-BEC) was synthesized following a procedure recently reported by Liang et al. using a dicationic piperidinium derivative OSDA (BPMPOH)[54]. The silica source is a colloidal silica suspension (Ludox AS-40), the germanium source is germanium oxide (99.999%, Merck), the aluminum source is aluminum hydroxide (65% $Al_2O_3$, Sigma-Aldrich) and the fluoride source is hydrofluoric acid (HF, 48%). The composition of the synthesis gel was: $0.71\ SiO_2$: $0.29\ GeO_2$: 0.05 $Al(OH)_3$: 0.25 BPMPOH: 0.5 HF: 7.5 $H_2O$. In a typical synthesis, 0.712 g $GeO_2$ was first mixed with an aqueous solution containing 11.85 wt% of BPMPOH and kept under stirring at room temperature until complete dissolution. Then, 0.184 g of aluminum hydroxide and 2.5 g of Ludox AS-40 were added under stirring. The mixture was kept under stirring at room temperature for 1 h until forming a homogeneous gel. The

resultant gel was heated at 50 °C to allow evaporation to reach the target $H_2O$ content. Finally, 0.465 g of the HF solution was added to the gel under stirring with a plastic spatula. The solid was transferred to a Teflon-lined stainless autoclave and heated in an oven at 150 °C under agitation for crystallization. After 48 h, the autoclave was cooled and the solid product was collected by filtration and washed by abundant distilled water. The product was dried in an oven at 100 °C in air overnight, and, afterwards, calcined in flowing air using a tubular furnace at 550 °C for 6 h with a heating rate of 1.5 °C/min.

**ITQ-33 (ITT).** ITQ-33 zeolite was synthesized following a previous report using a combination of hexamethonium hydroxide (MSPTOH) and hexamethonium bromide (HXMBr) as OSDA[55]. The silica source is TEOS (98%, Merck), the germanium source is germanium oxide (99.999%, Merck), the aluminum source is aluminum isopropoxide (Sigma-Aldrich) and the fluoride source is hydrofluoric acid (HF, 48%). The composition of the synthesis gel was: 0.67 $SiO_2$: 0.33 $GeO_2$: 0.05 $Al(OH)_3$: 0.15 $HXM(OH)_2$: 0.1 $HXMBr_2$: 0.3 HF: 2.5 $H_2O$. In a typical synthesis, 5.07 g of $GeO_2$ was first mixed with 66.49 g of an aqueous solution containing 18.8 wt% of $HXM(OH)_2$ and kept stirring at room temperature until complete dissolution. Then, 53.2 g of an aqueous solution of $HXMBr_2$ (20%), 1.51 g of aluminum isopropoxide, and 20.56 g of TEOS were added under stirring. The mixture was kept under stirring at room temperature for 16 h to allow full hydrolysis of TEOS. Afterwards, the gel was heated at 50 °C to allow evaporation of excessive $H_2O$ and ethanol generated from the hydrolysis of TEOS. When $H_2O$ content reached the desired value, 1.84 g HF was added to the gel under stirring with a plastic spatula. The solid was transferred to a Teflon-lined stainless autoclave and heated in an oven at 175 °C under static conditions for crystallization. After 24 h, the autoclave was cooled and the solid product was collected by filtration and washed by abundant distilled water. Then, the product was dried in an oven at 100 °C in air. The zeolite product was calcined in flowing air using a tubular furnace at 550 °C for 6 h with a heating rate of 1.5 °C/min.

**ITQ-27 (IWV).** ITQ-27 zeolite was synthesized following a procedure reported by Li et al. using diphenyldimethylphosphonium $DPDMP^+$ as OSDA[24]. In a typical synthesis, 30.01 g of a hydroxide solution of $DPDMP^+$ (8.12/wt in water) was mixed with 4.37 g of tetra-ethylorthosilicate (TEOS, Sigma-Aldrich, 98%wt) and 0.12 g of aluminum isopropoxide (IPA, Sigma-Aldrich, 98%wt). The mixture was stirred until the ethanol and isopropanol formed upon hydrolysis of TEOS and IPA were evaporated by heating at 50 °C. The weight of the synthesis gel was carefully weighed and the amount of water controlled to $H_2O:SiO_2 = 3.4$ to facilitate further adding of HF. Finally, 0.44 g of HF solution (SigmaAldrich, 48%wt in water) was added and stirred manually with a Teflon spatula, resulting in a thick gel. The final gel composition was $SiO_2$ / 0.028 Al / 0.5 DPDMP / 0.5 HF / 4 $H_2O$. This gel was transferred to a Teflon-lined stainless steel autoclave and heated at 150 °C in an oven under tumbling conditions. The solid was recovered by filtration, extensively washed with water, and dried at 100 °C overnight. The material was calcined at 580 °C for 6 h in air to remove the organic content located within the crystalline material.

## Characterization
Powder X-ray diffraction (PXRD) measurements were performed with a multisample Philips X'Pert diffractometer equipped with a graphite monochromator, operating at 40 kV and 35 mA, using Cu Kα radiation (λ = 0.1542 nm).

The chemical analyzes were carried out in a Varian 715-ES ICP-Optical Emission Spectrometer. The samples were dissolved in $HNO_3$/HCl/HF aqueous solution before measurement. The organic content of the as-synthesis materials was measured by elemental analysis performed with a SCHN FISONS elemental analyzer.

Textural properties, including BET surface area, micropore volume and external surface area of the samples, were measured by $N_2$ adsorption/desorption in a Micromeritics ASAP2000. The morphological feature and particle sizes were determined by field emission scanning electron microscope (FESEM, JEOL JSM-6300).

MAS NMR spectra were recorded in a Bruker AVANCE III HD 400 WB. $^{27}Al$ MAS NMR and $^{27}Al$ MAS NMR spectra were recorded in a Bruker 3.2 mm probe at a spinning rate of 20 kHz. $^{27}Al$ MAS NMR spectra were recorded with π/12 pulse length of 0.5 μs with a 1 s repetition time. $^{27}Al$ MAS NMR spectra were acquired with a selective zero quanto z-filter pulse sequence. $^{27}Al$ chemical shift was referred to $Al^{3+}(H_2O)_6$.

The acidity of the zeolites was determined by infrared spectroscopy combined with adsorption–desorption of pyridine at different temperatures. Infrared spectra were measured with a Nicolet 710 FT-IR spectrometer. Pyridine adsorption–desorption experiments were carried out on selfsupported wafers (10 mg/cm²) of original samples previously activated at 400 °C and $10^{-2}$ Pa for 2 h. After wafer activation, the base spectrum was recorded and pyridine vapor ($6.5 \times 10^2$ Pa) was admitted into the vacuum IR cell and adsorbed onto the zeolite. Desorption of pyridine was performed under vacuum over 1 h periods of heating at 350 °C, followed by IR measurement at room temperature. All the spectra were scaled according to the sample weight. The numbers of Brønsted and Lewis acid sites were determined from the intensities of the bands at ca. 1545 and 1450 cm$^{-1}$, respectively.

## Reaction
Prior to catalytic tests, all zeolites were pelletized, crushed and sieved to a particle size of between 0.2 to 0.4 mm. Alternatively, pellets with particle sizes of between 0.1 to 0.2 mm and 0.4 to 0.8 mm were also prepared to investigate to intraparticle diffusion. In all cases, particle sizes of 0.1 to 0.2 mm and 0.2 to 0.4 mm gave the same catalytic results while 0.4 to 0.8 mm gave slightly lower activity, indicating that intraparticle diffusion starts to appear when pellet size approximates 0.8 mm. Then, the pellet size of between 0.2 to 0.4 mm was selected for all catalytic tests.

The gas phase transalkylation of diethylbenzene with benzene was carried out in a fixed bed reactor equipped with a bypass line in parallel with the reactor. The feedstock is a liquid mixture of benzene and diethylbenzene (Bz:DEB weight ratio 3:1) containing n-octane as internal standard. The required amount of pelletized catalyst was properly mixed with silicon carbide to reach a total volume of 12.6 ml. The catalyst was activated in dry $N_2$ flow at 540 °C for 3 h and then the temperature of the reactor was cooled down to the reaction temperature. The abovementioned liquid mixture was fed into the bypass together with the desired amount of $N_2$ as carrier gas. The composition of the feeding gas had a molar ratio of $N_2$: DEB: Bz = 30: 1: 5 and was monitored on-line with a Varian-3800 gas chromatograph equipped with a 30 m 5% phenyl / 95% dimethyl polysiloxane capillary column connected to a flame ionization detector. At this stage, the sample from outlet of the reactor was analyzed every 7 min. When the outlet composition according to GC was stable, the feeding was switched to the reactor line, and the outlet of the reactor was analyzed by the same gas chromatograph every 9 min. The initial activity was measured by precisely sampling at Time On Stream = 0 s.

Prior to catalytic test, the absence of diffusion limitation was evaluated by testing the reaction under the abovementioned procedure changing feeding flow. In gas phase reaction, the $N_2$ flow was adjusted within the range of 25 ml/min to 1000 ml/min while maintaining the partial pressure of the aromatics. When the flow was above 125 ml/min, the reactions gave identical catalytic results. Then all the gas phase reactions were conducted with a minimum $N_2$ flow of 250 ml/min to avoid diffusion limitation.

## Kinetic study

Transalkylation reaction between Bz and DEB is a bimolecular reaction. Since benzene is fed in an excessive amounts than stoichiometric ratio, the concentration of Bz could be considered constant and the reaction as first-order reaction. Then, the first-order rate law equation was employed to obtain the rate constant:

$$k = \frac{\ln(1 - Conv.DEB)}{w/F} = \frac{\ln(1 - \frac{1}{2} \cdot Yield.EB.)}{w/F} \qquad (1)$$

Where k is the rate constant, and w/F is the contact time.

Activation energy (Ea) was calculated using the Arrhenius equation, by plotting lnk against 1/T:

$$lnk = \frac{-E_a}{R} \cdot \frac{1}{T} + \ln A \qquad (2)$$

where k is the reaction rate constant, T is the reaction temperature and A is the pre-exponential factor.

## Data availability

The experimental and computational data that support the findings of this study are available from the corresponding author upon reasonable request.

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

## Acknowledgements

This work has been supported by Spanish Government through CEX2021-001230-S, PID2020-112590GB-C21, PID2021-122755OB-I00 and TED2021-130739B-I00 (MCIN/AEI/FEDER, UE), by Generalitat Valenciana through AICO/2021/201, and CSIC through the I-link+ Program (LINKA20381). The authors thank Dr. María José Díaz-Cabañas and Dr. Pilar Cumplido for providing the ITQ-24 and ITQ-15 samples. The Electron Microscopy Service of the UPV is acknowledged for their help in sample characterization. Red Española de Supercomputación (RES) and Servei d'Informàtica de la Universitat de València (SIUV) are acknowledged for computational resources and technical support. P.F. and C.Li. thank ITQ for their contract. M. Xie thanks the Agency for Science, Technology and Research for funding. D Schwalbe-Koda acknowledges financial support from MIT Energy Initiative Fellowship. MIT Engaging cluster, and MIT Lincoln Lab Supercloud cluster at MGHPCC are gratefully acknowledged for computational resources and support. The authors thank the MIT-Spain INDITEX Circularity Seed Fund for financial support.

## Author contributions

A.C. conceived and directed the project. M.B. designed and directed the computational work. P.F. carried out the DFT calculations. M.M. directed the zeolite synthesis work. C.L. performed the zeolite synthesis and catalytic experiments. M.X. and D.S.K. performed the FF screening supervised by R.G.B., M.B., M.M., and A.C. wrote the manuscript. All authors participated in the discussion and interpretation of the results and contributed to the manuscript preparation and revision.

## Competing interests

The authors declare no competing interests.
