## [Peer Review File · Nature Communications]

Approaching enzymatic catalysis with zeolites or how to select one reaction mechanism when competing with othersREVIEWER COMMENTS

Reviewer #1 (Remarks to the Author):

This paper provides a computational methodology for the “in silico” selection of the most adequate existing zeolites as catalysts for the competitive transalkylation and disproportionation of diethylbenzene reaction. The methodology combines a fast screening of potentially efficient zeolites and the comparison of binding energy and proton shift energy barrier of competitive transalkylation and disproportionation of diethylbenzene reactions on the most promising catalyst candidates based on the DFT calculations. The approach could help to select the most efficient catalyst for a given reaction with different competitive routes catalyzed by zeolites. This work could be accepted for publication in Nature Communication after the following points are fully taken into account.

1. The schemes of the mechanism for the DEB disproportionation should be provided in the manuscript.
2. The proton shift steps are computed on the pure silica zeolites with periodic models. However, the interactions among the net positive charges (on the intermediates in the pure silica zeolite) in periodic models may cause the error in the energy calculations. Thus, the authors should confirm that the net positive charges in the periodic models do not affect the energy calculations.
3. The energy barriers of the proton shift steps are compared in part 2.2. However, compared with the proton shift steps, the alkyl-transfer step of the two routes (transalkylation and the disproportionation) may be more sensitive to the zeolite confinement, thus the reaction barriers and transition states of the alkyl-transfer step for the DEB transalkylation and disproportionation routes should be computed and compared in the manuscript.

Reviewer #2 (Remarks to the Author):

The authors describe a new method for designing zeolite catalysts as enzyme-like catalysts on transalkylation and disproportionation of diethylbenzene competing on zeolite catalysts. First, the authors narrowed down the zeolite structures based on the binding energy (BE) of OSDA having a structure similar to that of the reaction intermediate. The zeolite structures were further narrowed down based on the knowledge of activity-structure (pore size and channel system) relationship for the transalkylation reaction of diethylbenzene over zeolite catalysts. For the candidate zeolite structures, the BE of the intermediates of the two competing reactions, i.e., BE of I_trans and BE of I_disp, were evaluated as the descriptors of the activity and selectivity of the reaction. In addition, the activation energies (Ea) of the transalkylation and the disproportionation reactions were also evaluated as other descriptors. Based on the descriptors and the experimental reaction results, the authors concluded that the ratio of BE of intermediates relates to the selectivity, and the transition state energies of the intermediates relates to the reaction rate. These results indicate that structural control of zeolite can control the BE and the transition state energies of the intermediates and thus achieve enzyme-like selective reactions. The results of this study will be of interest for the development of zeolite catalysts of great practical importance. However, there appear to be logical or scientific flaws. I would suggest that

the authors address the following comments:

1. On page 8, “Despite the differences in methodology and model, a good match between the FF and DFT energetics is observed...”: What is the criterion for judging whether the match is good or not? The FF calculation results of BEA, IWW, and MOR do not show less than 1 of $I_{\text{disp}}/I_{\text{trans}}$.
2. In Fig. 7(a), IWR is out of the trend. The authors explained that this is due to the overstabilization of I_{disp} at the intersection between 12R and the two 10R channels causing slow diffusion of TBE. What is the overstabilization? How is it different from the simple stabilization described by the BE of I_{disp} ? Please explain more about the overstabilization.
3. In Fig. 7(b), there seems to be a trend that r_{trans} increases with BE I_{trans} . But, the authors concluded that there is no correlation between them. The authors should give a convincing explanation for this conclusion.
4. In Fig. 8, only six zeolites are evaluated, while eight zeolites are evaluated in Fig. 7. Why did the authors exclude FAU and ITT in Fig. 8? After adding the data of FUA and ITT in Fig. 8, the correlation between $E_{\text{a_exp}}$ and $E_{\text{a3_DFT}}$ should be compared to that between r_{trans} and BE I_{trans} to support their conclusion that E_{a} should be calculated to predict the catalytic activity (reaction rate).

Minor comments:

1. On page 14, “...the planarity of the two aromatic...” might be “...the planarity of the two aromatic...”.
2. It might be better to explain dotted lines in Figure 1.
3. Table 1: What is “kJ/mol l”?
4. Figure 7(b): What is “kJ/mol l”? It might be “kJ/mol”.

POINT BY POINT RESPONSE TO REVIEWERS COMMENTS

Reviewer #1:

This paper provides a computational methodology for the “in silico” selection of the most adequate existing zeolites as catalysts for the competitive transalkylation and disproportionation of diethylbenzene reaction. The methodology combines a fast screening of potentially efficient zeolites and the comparison of binding energy and proton shift energy barrier of competitive transalkylation and disproportionation of diethylbenzene reactions on the most promising catalyst candidates based on the DFT calculations. The approach could help to select the most efficient catalyst for a given reaction with different competitive routes catalyzed by zeolites. This work could be accepted for publication in Nature Communication after the following points are fully taken into account.

1. The schemes of the mechanism for the DEB disproportionation should be provided in the manuscript.

Response: Following the Reviewer’s recommendation we have included the mechanism for DEB disproportionation in the revised manuscript. For clarity, the original Scheme 2 has been split into two Schemes. The new Scheme 2 includes the alkyl-transfer pathway for the three possible competing processes: transalkylation, disproportionation and dealkylation. The new Scheme 3 includes the diaryl-mediated pathway for the competing transalkylation and disproportionation, together with the OSDA mimic and the key diaryl intermediates.

Scheme 2. Alkyl-transfer pathway for transalkylation of diethylbenzene with benzene and competitive dealkylation yielding ethene and disproportionation producing triethylbenzene.

Scheme 3. (a) Diaryl-mediated pathway for the transalkylation of diethylbenzene with benzene and for the competing disproportionation yielding triethylbenzene represented by the red bonds in all structures. (b) Structures of the OSDA mimic and of the diaryl intermediates for transalkylation (I_{trans}) and disproportionation (I_{disp}).

2. The proton shift steps are computed on the pure silica zeolites with periodic models. However, the interactions among the net positive charges (on the intermediates in the pure silica zeolite) in periodic models may cause the error in the energy calculations. Thus, the authors should confirm that the net positive charges in the periodic models do not affect the energy calculations.

Response: The Reviewer raises here an important point regarding the reliability of periodic calculations for charged systems, since it is true that the interactions among the net charges in periodically repeated cells might lead to wrong energies. To confirm this is not the case we have used more realistic Al-containing neutral models in which the positive charge on the organic species is compensating the negative charge generated by the substitution of one framework Si atom with an Al atom. Then, we have calculated the

energy profiles for the four diaryl-mediated pathways considered in this study and we have compared the energy profiles obtained with the charged (pure Si framework) and with the neutral (Al-containing framework) models.

The results for IWV and MOR structures were taken from a previous study in our group, published in *J. Am. Chem. Soc.* 2021, 143, 10718–10726, and we have now expanded this study to UTL containing larger 14-ring channels and to BOG containing narrower 10-ring channels, to be sure that the results reported are correct irrespectively of the size of the channels. Al positions for each zeolite were chosen by intrinsic stability criteria placing one unique Al per unit cell. The most stable locations for Al are T1 in BOG, T3 closely followed by T6 in IWV, T4 in MOR and T11 in UTL, as summarized in new Table S5 in the revised Supplementary Information. Since the T11 site in UTL is at the intersection between the 12-ring and the 14-ring channels, the two possible locations considered in the original manuscript for the mechanistic study, UTL(cha) and UTL(int) were also included in the Al-containing models.

The comparative free energy profiles depicted in the new Supplementary Figures 4-7, with parallel orange and blue profiles, and the relative energies of all minima and transition states summarized in new Supplementary Table 6 in the revised Supplementary Information show that the calculations with the charged models discussed in the original manuscript are not wrong or biased due to interactions between positive charges. As indicated by the Reviewer, this information is relevant, and therefore it has been included in page 12 of the revised manuscript: *“In order to discard any deviations in the calculated energies associated to interactions between net positive charges in periodically repeated cells, a comparison with previous²⁴ and new data obtained with Al-containing neutral models is presented in Supplementary Tables 5-6 and Supplementary Figures 4-7. The comparison confirms the accuracy of the approach followed.”*

The new Tables and Figures follow:

Supplementary Table 5. Relative stability of Al location in BOG and UTL zeolite structures.

BOG	Erel (kJ/mol)	UTL	Erel (kJ/mol)
T1	0	T1	35
T2	23	T2	41
T3	35	T3	16
T4	24	T4	11
T5	6	T5	28
T6	6	T6	39
		T7	37
		T8	20
		T9	19
		T10	43
		T11	0

Supplementary Table 6. Relative stability (in kJ/mol) of intermediates and transition states involved in the mechanisms of transalkylation in pure silica and Al-containing models of BOG, IWV, MOR and UTL zeolite structures.

	BOG		IWV			MOR		UTL (int)		UTL (cha)	
	Si	Al-T1	Si	Al-T3	Al-T6	Si	Al-T4	Si	Al-T11	Si	Al-T11
I1 ⁺	0	0	0	0	0	0	0	0	0	0	0
TS1	75	74	56	68	68	95	95	59	70	64	74
I4 ⁺	-17	-24	-25	-19	-30	-25	-25	-27	-11	-30	-7
TS2	29	39	21	32	5	33	58	29	43	28	29
I2 ⁺	-20	-27	-34	-8	-12	-21	-20	-33	-31	-27	-16
TS3	45	29	18	28	28	57	57	16	23	30	50
I3 ⁺	-28	-21	-18	-16	-18	-19	-18	-32	-21	-22	0
TS4	22	32	29	26	21	31	32	17	28	23	50
TS5	88	74	60	76	43	84	113	82	93	68	93
TS6	76	79	52	77	68	66	86	72	69	53	68

Supplementary Figure 4. Calculated energy profiles for all transalkylation diaryl-mediated pathways in pure Si (blue) and Al-containing (orange) models of BOG.

Supplementary Figure 5. Calculated energy profiles for all transalkylation diaryl-mediated pathways in pure Si (blue) and Al-containing (orange and yellow) models of IWV. Data taken from reference 30.

Supplementary Figure 6. Calculated energy profiles for all transalkylation diaryl-mediated pathways in pure Si (blue) and Al-containing (orange) models of MOR. Data from reference 30.

Supplementary Figure 7. Calculated energy profiles for all transalkylation diaryl-mediated pathways in pure Si (blue) and Al-containing (orange) models of UTL(cha) (full lines) and UTL(int) (dashed lines).

3. The energy barriers of the proton shift steps are compared in part 2.2. However, compared with the proton shift steps, the alkyl-transfer step of the two routes (transalkylation and the disproportionation) may be more sensitive to the zeolite confinement, thus the reaction barriers and transition states of the alkyl-transfer step for the DEB transalkylation and disproportionation routes should be computed and compared in the manuscript.

Response: In a previous work (JACS 2021, 143, 10718) we studied extensively the two possible pathways (alkyl-transfer and diaryl-mediated) in two zeolite structures, MOR and IWV. We found that the carbonium ion intermediates involved in the alkyl-transfer pathway are quite unstable, and that the transfer of a proton or an ethyl group between the organic fragment and the zeolite framework imply higher activation energies than the proton shifts in the diaryl-mediated pathway. However, as pointed by the Reviewer, these energies might be more sensitive to the zeolite confinement and following his/her recommendation we have now calculated this pathway also in BOG and UTL zeolites, which produce experimentally the highest (4,8%) and lowest (0%) amount of ethene, respectively. These results are summarized in new Supplementary Table 9 and new Supplementary Figure 8, and the following paragraph discussing these data has been included in page 17 of the revised manuscript: *“Finally, the complete mechanism for the transalkylation reaction following the alkyl-transfer route was calculated in BOG and UTL and compared with results published previously for MOR and IWV²⁴ in order to confirm that this pathway is not competitive in the zeolite structures selected to stabilize diaryl intermediates. The values in Supplementary Table S9 and the profiles depicted in Supplementary Fig. 8 confirm that the ethyl transfer between the zeolite framework and the organic fragment is the most energetically demanding step, with TS8 and TS9 being between 135 and 160 kJ/mol higher in energy than the initial. Such high activation energies indicate that the contribution of the alkyl-transfer pathway to DEB conversion should be low or negligible in the selected zeolite structures.”*

The new Tables and Figures follow:

Supplementary Table 9. Relative stability (in kJ/mol) of intermediates and transition states involved in the mechanisms of transalkylation in pure silica and Al-containing models of BOG, IWV, MOR and UTL zeolite structures. The data for IWV and MOR are taken from reference 30.

	BOG		IWV		MOR	UTL	
	10R	12R	T3	T6	T4	12R	14R
ZH+ DEB + Bz	0	0	0	0	0	0	0
TS7	77	82	48	57	84	99	107
Z ⁻ + DEBH ⁺ +Bz	56	33	44	53	54	30	65
TS8	152	127	137	155	135	145	129
Z-Et + EB + Bz	69	45	78	88	71	49	30
TS9	160	135	148	142	161	159	142
Z ⁻ + EBH ⁺ +EB	89	63	121	108	80	69	93
TS10	97	80	137	120	98	124	115
ZH + 2 EB	27	-4	25	-2	-19	8	0

Supplementary Figure 8. Calculated energy profiles for the alkyl-transfer pathway for transalkylation in BOG, IWV, MOR and UTL zeolite structures. The data for IWV and MOR are taken from reference 30.

Following the Reviewer’s recommendation we have also calculated the complete alkyl-transfer mechanism for the disproportionation reaction in BOG, IWV, MOR and UTL zeolites. The relative energies of all intermediates and transition states involved are summarized in Table R1 below. However, since the alkyl-transfer pathway is not competitive for the transalkylation reaction and the disproportionation reaction is in all cases more energetically demanding, these data have not been included in the revised manuscript to avoid making it unnecessarily long.

Table R1. Relative stability (in kJ/mol) of intermediates and transition states involved in the mechanisms of disproportionation in pure silica and Al-containing models of BOG, IWV, MOR and UTL zeolite structures.

	BOG		IWV		MOR	UTL	
	10R	12R	T3	T6	T4	12R	14R
ZH+ DEB + Bz	0	0	0	0	0	0	0
TS7	51	95	73	98	62	107	86
Z- + DEBH+ +Bz	46	58	62	89	43	29	80
TS8	180	134	136	170	143	153	148
Z-Et + EB + Bz	91	51	58	103	19	37	54
TS9	166	132	180	164	165	222	168
Z- + EBH+ +EB	205	35	83	100	33	49	95
TS10	431	135	91	224	64	107	104
ZH + 2 EB	148	-31	0	33	-13	80	23

Reviewer #2:

The authors describe a new method for designing zeolite catalysts as enzyme-like catalysts on transalkylation and disproportionation of diethylbenzene competing on zeolite catalysts. First, the authors narrowed down the zeolite structures based on the binding energy (BE) of OSDA having a structure similar to that of the reaction intermediate. The zeolite structures were further narrowed down based on the knowledge of activity-structure (pore size and channel system) relationship for the transalkylation reaction of diethylbenzene over zeolite catalysts. For the candidate zeolite structures, the BE of the intermediates of the two competing reactions, i.e., BE of I_{trans} and BE of I_{disp}, were evaluated as the descriptors of the activity and selectivity of the reaction. In addition, the activation energies (E_a) of the transalkylation and the disproportionation reactions were also evaluated as other descriptors. Based on the descriptors and the experimental reaction results, the authors concluded that the ratio of BE of intermediates relates to the selectivity, and the transition state energies of the intermediates relates to the reaction rate. These results indicate that structural control of zeolite can control the BE and the transition state energies of the intermediates and thus achieve enzyme-like selective reactions. The results of this study will be of interest for the development of zeolite catalysts of great practical importance. However, there appear to be logical or scientific flaws. I would suggest that the authors address the following comments:

1. On page 8, “Despite the differences in methodology and model, a good match between the FF and DFT energetics is observed...”: What is the criterion for judging whether the match is good or not? The FF calculation results of BEA, IWW, and MOR do not show less than 1 of I_{disp}/I_{trans}.

Response: We agree with the Reviewer that the sentence was not adequate, since we cannot claim a good match between the FF and DFT computed values. Instead, what is shown in Figure 1b is that the trend provided by the two methodologies is in good agreement: all candidates are able to stabilize the diaryl intermediates, and increasing BE I_{trans} also increases BE I_{disp}. The text in page 9 of the revised manuscript has been modified as follows: “*Despite the differences in methodology and model, the FF and DFT values follow the same relationship in the plot of BE I_{disp} versus BE I_{trans} in Figure 1b, thus confirming the validity of the trend proposed by the computationally cheaper and faster FF calculations.*”

2. In Fig. 7(a), IWR is out of the trend. The authors explained that this is due to the over-stabilization of I_{disp} at the intersection between 12R and the two 10R channels causing slow diffusion of TBE. What is the over-stabilization? How is it different from the simple stabilization described by the BE of I_{disp}? Please explain more about the over-stabilization.

Response: We thank the Reviewer for noticing this inadequate use of the word overstabilization, which has been corrected in the revised manuscript. The reason for IWR being out of the trend in Fig 7(a) is that the I_{disp} intermediate is very stable at the intersection between the 12-ring and 10-ring channels, and such high stability makes the diffusion of TEB slow. The sentence in page 19 of the revised manuscript has been modified as follows: *“In contrast, less TEB than expected from the theoretical BE ratios I_{disp}/I_{trans} is experimentally detected in IWR. The reason is the tight fitting of the diaryl intermediate for disproportionation I_{disp} at the intersection between the 12-ring and the two 10-ring channels. At this specific position the two ethyl groups in one of the aromatic rings fit perfectly into the two narrow 10-ring channels of IWR leading to an excellent stabilization of I_{disp} but hindering the diffusion of TEB out of the crystal.”*

3. In Fig. 7(b), there seems to be a trend that r_{trans} increases with BE I_{trans} . But, the authors concluded that there is no correlation between them. The authors should give a convincing explanation for this conclusion.

Response: As indicated by the Reviewer, there seems to be a trend that r_{trans} increases with BE I_{trans} , but the trend is not fully consistent along the dataset and the quantitative correlation is not good. For instance, two of the three zeolites with the largest BE I_{trans} , MOR and BOG, exhibit the lowest reaction rates, ~ 300 mol_{EB}/mol_{acid} h, while the most active catalyst, IWV, is the one with the lowest BE, which is opposite to expected. There are three zeolites with similar reaction rates of ~ 600 - 700 mol_{EB}/mol_{acid} h with BE I_{trans} ranging from -145 (the strongest BE) to -110 kJ/mol (quite weak) highlighted in yellow Figure R1. And there are three zeolites with nearly the same BE I_{trans} , between -102 and -105 kJ/mol, with reaction rates ranging from 1000 to 2000 mol_{EB}/mol_{acid} h, highlighted in pink in Figure R1. Altogether, there is not a clear relationship between BE and reaction rate. Taking to account the Reviewer's comment, the sentence in page 21 has been modified as follows: *“On the other hand, there seems to be a rough trend in the plot of the experimental reaction rates for transalkylation r_{trans} and the calculated BE for I_{trans} , with higher reaction rates in the zeolite structures that stabilize less the I_{trans} intermediates (Figure 7b), but the trend is not consistent along the dataset and the correlation is not good.”*

Figure R1. Correlation between experimental reaction rates for transalkylation r_{trans} and DFT calculated BE of the diaryl I_{trans} intermediate.

4. In Fig. 8, only six zeolites are evaluated, while eight zeolites are evaluated in Fig. 7. Why did the authors exclude FAU and ITT in Fig. 8? After adding the data of FAU and ITT in Fig. 8, the correlation between Ea_{exp} and Ea_{3_DFT} should be compared to that between r_{trans} and $BE I_{trans}$ to support their conclusion that Ea should be calculated to predict the catalytic activity (reaction rate).

Response: In Figure 7 the experimentally measured reaction rates are plotted against the binding energy of the diaryl intermediate for transalkylation $BE I_{trans}$, which was calculated for the twelve zeolite structures listed in Table 1, including both ITT and FAU. However, in section 2.2, the mechanistic study was carried out only on six selected zeolites, as indicated in page 12: “*To analyse the influence of the zeolite framework on the kinetics of the transalkylation and disproportionation reactions, the four pathways described above were investigated in pure silica models of the promising BEC, BOG, IWR, IWV and UTL zeolites using periodic DFT calculations. This selection includes structures with 2D and 3D channels systems containing 10-, 12-, and 14-ring channels, all of them with good BE for the neutral transalkylation I_{trans} intermediate. MOR, with a 1D channel system where the diaryl intermediates do not match too well, was included for comparison.*” ITT was excluded because the calculated I_{disp}/I_{trans} ratio in Table 1 was the largest, suggesting an important contribution of the disproportionation reaction that was confirmed experimentally by the results in Table 3. FAU was excluded for practical computational reasons. The calculated I_{disp}/I_{trans} ratio for FAU in Table 1 is similar to that of BOG, but the FAU unit cell is much larger than any other zeolite considered in this

work. It contains 192 T atoms as compared to 96 T atoms for the Orthorhombic BOG unit cell, making the mechanistic study too computationally demanding in FAU.

In any case, as explained in the previous point raised by the Reviewer, the rough trend in the plot of reaction rate r_{trans} and BE I_{trans} is opposite to expected, with the most active zeolites being those that stabilize less the diaryl I_{trans} intermediate. In contrast, the correlation between experimental and calculated activation energies shown in Figure 8 is direct, and the experimentally determined reaction rates r_{trans} increase as the calculated activation energies decrease, following the expected relationship according to kinetics.

Minor comments:

1. On page 14, "...the planarity of the two aromatic..." might be "...the planarity of the two aromatic...".

Response: Thanks for noticing, it was a typo and it has been corrected in the revised manuscript.

2. It might be better to explain dotted lines in Figure 1.

Response: As suggested by the Reviewer we have indicated in the caption of Figure 1 the meaning of the dotted lines:

Figure 1. Correlations between BE I_{trans} and BE I_{disp} for different types of zeolite channels system calculated with FF (a) and comparison between FF and DFT for the selected candidates (b). *The dotted lines corresponding to $BE I_{\text{disp}} = BE I_{\text{trans}}$ are included to guide the eye.*

3. Table 1: What is "kJ/mol I"?

Response: As indicated in page 9 of the revised manuscript, to calculate the binding energies using DFT "*the zeolite pores were filled with the maximum number of molecules possible, given in column n in Table 1.*" The BE I_{trans} and BE I_{disp} given in Table 1 are "*binding energies per mole of diaryl intermediate*" and this is why they were given as kJ/mol I. However, we agree with the Reviewer that the units are kJ/mol and it has been corrected in the revised manuscript.

4. Figure 7(b): What is "kJ/mol I"? It might be "kJ/mol".

Response: The answer to this point is the same as previous one. Following the Reviewer's recommendation the units in Figure 7(b) have been corrected.

REVIEWERS' COMMENTS

Reviewer #1 (Remarks to the Author):

Since the authors have fully addressed my concerns and made the corresponding modifications, I recommend to accept the revised manuscript for publication in Nature Communication as is

Reviewer #2 (Remarks to the Author):

The authors addressed all my concerns. I recommend the publication of the revised manuscript.